# Mass casualty incident preparedness and response: A desk review of the Code Orange Plan and Assessment of Healthcare Workers' Knowledge, Attitudes, and Practices in a Lebanese Tertiary Government Hospital

**Linda Abou-Abbas**[1]ʘ*, **Rima Kashash**[2]ʘ, **Mustapha Khalife**[1], **Mohamad Shafic Ramadan**[1]

**1** International Committee of the Red Cross, Beirut, Lebanon, **2** Faculty of Health Sciences, American University of Beirut, Beirut, Lebanon

ʘ Contributed equally as first co-authors
* labouabbas@icrc.org

## Abstract

### Background

Effective preparedness and response to mass casualty incidents (MCI) are essential for hospital safety, operational efficiency, and the delivery of timely, high-quality patient care during emergencies. This study assessed a tertiary government hospital in Lebanon's Code Orange plan by reviewing documentation for alignment with international guidelines and evaluating staff knowledge, attitudes, and practices (KAP) regarding MCI preparedness.

### Methods

Documents reviewed at Rafik Hariri University Hospital (RHUH) included the current Code Orange plan, relevant policies, and international guidelines. A comprehensive evaluation framework was used, focusing on preparedness, incident command systems, communication, and management. A comparison with established standards was conducted to identify gaps. Complementing this, a cross-sectional study was conducted using a convenient sample of medical and non-medical healthcare workers to evaluate their KAP regarding MCI preparedness.

### Results

The desk review of the RHUH Code Orange plan identified both strengths and significant gaps in MCI preparedness. While the plan defines staff roles and resources for emergency response, it lacks detailed procedures for activation strategies, surge capacity, continuity of essential services, and triage processes. Additionally, post-event recovery protocols are insufficient or absent, and the importance of regular

**Data availability statement:** The data supporting the findings of this study are owned by the International Committee of the Red Cross (ICRC) and are subject to legal and ethical restrictions related to participant confidentiality. The dataset cannot be made publicly available. Access to the data may be granted to qualified researchers for replication or secondary analysis purposes, subject to approval by the ICRC and in accordance with its data protection and ethical policies. Data access requests should be directed to the ICRC through its Geneva Core mailbox at gva_core_mailbox@icrc.org.

**Funding:** This study was funded by the Agence Française de Développement (AFD) as part of its partnership project with the International Committee of the Red Cross (ICRC) and Rafik Hariri University Hospital (RHUH). AFD had no role in the conceptualization, design, data collection, analysis, decision to publish, or preparation of the manuscript.

**Competing interests:** The authors have declared that no competing interests exist.

**Abbreviations:** MCI, Mass casualty incident; LMICs, Low and Middle-income countries; EMS, Emergency Medical Services; RHUH, Rafik Hariri University Hospital; ICRC, International Committee of the Red Cross; KAP, Knowledge, Attitude and practice; WHO, World Health Organization; CDC, Centers for Disease Control and Prevention; NIMS, National Incident Management System; ICS, Incident Command System; APIC, Association for Professionals in Infection Control and Epidemiology; HCW, Health care worker; SD, Standard deviation; IT, Information Technology; ER, Emergency Room

drills is not adequately emphasized. The KAP study revealed significant differences between medical and non-medical staff in terms of MCI knowledge, involvement, and training engagement, with medical staff reporting higher levels of familiarity and desire for participation.

## Conclusion

The findings underscore the need to bridge knowledge and engagement gaps between medical and non-medical staff to enhance MCI response. Key actions include interdisciplinary training to build coordination, clear communication protocols to streamline information flow, and routine drills with defined roles to strengthen preparedness. Additionally, implementing performance monitoring during drills and real MCIs, along with conducting regular evaluations, will allow for continuous refinement of response strategies.

---

## 1. Introduction

A mass casualty incident (MCI) occurs when a sudden influx of patients overwhelms healthcare systems, requiring immediate support [1,2]. MCI can result from a wide range of events, including hydro-meteorological disasters, transportation accidents, terrorism, armed conflicts, and Chemical, Biological, Radiological, Nuclear, and Explosive (CBRNE) incidents [3–5]. MCIs are inherently unpredictable and vary in nature, but they can still be foreseen and prepared for. Unlike pandemics, which start with a slow rise in cases that escalates over time, MCIs result in a rapid surge of casualties in a short timeframe [3].

During MCIs, hospitals play a critical role in providing essential medical care to their communities [4]. These incidents typically require a coordinated multijurisdictional and multifunctional response, with healthcare services at the core of recovery efforts. However, hospitals often struggle to maintain operations due to the surge in demand, limited resources, and disrupted communication and supply chains [6]. This challenge is particularly acute in low- and middle-income countries (LMICs), where underdeveloped healthcare infrastructure and a higher disease burden further hinder the response and recovery process [7]. Even in developed nations, hospitals face significant short-term challenges during an MCI, underscoring the importance of robust emergency planning to safeguard both staff and patients while ensuring continuity of care [8–11].

The role of emergency medical services (EMS) in managing MCI is pivotal within the broader healthcare system [9]. Key tasks include managing the incident scene, conducting triage, administering life-saving interventions (e.g., airway management, bleeding control), providing rapid diagnosis and treatment, and transporting the injured to healthcare facilities. The EMS response to MCI generally follows a structured process, beginning when the telecommunication unit receives notification of the incident, triggering the activation of the EMS response system. Depending on the scale of the incident, other emergency services—such

as the police and fire departments—are also dispatched to the scene. Upon arrival, the first EMS teams establish a command post near the staging area, ensuring the safety and security of the scene before beginning triage, treatment, and patient transport under the direction of the operation command. Simultaneously, the emergency operations center (EOC) is activated to coordinate the response and request additional resources if needed. Hospitals are informed by the dispatch unit to activate their MCI response plans, ensuring that they are prepared to receive and manage the influx of patients [1,4,12].

Efficient communication is crucial throughout the operation, with the EMS teams at the scene maintaining a dedicated communication link with mobile dispatch. The operation continues until all casualties have been treated and transported to hospitals. Throughout this process, the coordination between the incident command, EMS teams, and hospitals is facilitated through the central dispatch, ensuring that all elements of the response are integrated and working toward a unified goal [4].

Studies reveal significant gaps in MCI preparedness, including confusion over roles and responsibilities, poor communication, lack of planning, inadequate training, and outdated plans [13–16]. Given the diverse nature of MCI and the requirement for short-term, high-intensity management, it is crucial to develop a specialized preparedness framework tailored specifically to MCI [4]. This framework must first identify the essential components and elements of EMS preparedness, ensuring that the system can respond effectively to a wide range of potential MCI scenarios. Such a framework should focus on building capacity in critical areas like communication, inter-agency coordination, rapid resource mobilization, and continuous training to ensure that EMS teams can respond effectively under pressure, mitigate casualties, and maintain system integrity during high-stakes incidents [17–19].

Rafik Hariri University Hospital (RHUH), the largest governmental university hospital in Lebanon, serves as a vital safety net for vulnerable populations of all nationalities [20]. As a key first responder for MCIs and disasters in the country [18,21–23], RHUH plays a central role in the national healthcare system. In recognition of this role, the International Committee of the Red Cross (ICRC) established a long-term partnership with RHUH in 2016 to ensure access to quality care for vulnerable patients, strengthen hospital infrastructure, and build the capacities of RHUH staff [24]. This partnership has significantly enhanced RHUH's ability to provide essential healthcare services, particularly during emergencies.

To strengthen preparedness for MCIs, RHUH established its mass casualty response plan, Code Orange, in 2016. The plan provides a structured framework for hospital response during large-scale emergencies, including activation criteria, triage processes, staff mobilization, and surge capacity management. RHUH has applied this plan in response to several large-scale events, most notably the Beirut Port explosion in 2020, during which approximately 220 casualties presented to the hospital, with 48 requiring admissions, including critical care. This event placed unprecedented strain on the hospital and provided critical insights into both the strengths and limitations of existing response mechanisms.

Following the Beirut Port explosion, a formal debriefing was conducted to evaluate the hospital's response and inform updates to the Code Orange plan. Lessons learned from this real-world experience—particularly challenges related to rapid patient influx, interdepartmental coordination, and resource limitations—were translated into targeted improvements in triage systems, surge capacity, and staff mobilization protocols. These updates strengthened the plan's practicality and ensured that it is better aligned with the realities of large-scale MCIs.

Building on these efforts, a desk review was conducted jointly by the ICRC and RHUH to further update and strengthen existing response strategies. However, effective disaster preparedness extends beyond protocol development and requires a thorough understanding of staff readiness. Assessing healthcare workers' preparedness is essential to identify existing competencies, uncover skill gaps, and determine training needs.

Therefore, this study was conducted to address both structural and human components of MCI preparedness, with the following objectives:

1. To evaluate RHUH's mass casualty preparedness by conducting a detailed desk review of the Code Orange plan to identify areas for improvement and alignment with international guidelines.

2. To assess the knowledge, attitudes, and practices (KAP) as well as the perceptions of RHUH hospital staff regarding their preparedness for MCI to identify competency gaps and training needs.

3. To provide actionable recommendations for enhancing RHUH's mass casualty response plan and staff training programs.

## 2. Materials and methods

This project consisted of both a desk review and a cross-sectional study. The primary aim of the desk review was to evaluate the existing Code Orange plan at RHUH, focusing on identifying gaps, weaknesses, and areas for improvement to ensure that the plan is up-to-date, applicable, and aligned with practices and international guidelines in MCI preparedness. The desk review was further supplemented by a KAP study to assess the knowledge, attitudes, practices, and perceptions of RHUH hospital staff regarding MCI preparedness and response. This additional assessment aimed at enhancing the understanding of RHUH staff's awareness and implementation of MCI protocols, complementing the findings from the document review and providing valuable insights into the practical aspects of emergency preparedness.

### 2.1. Desk review methodology

**2.1.1. Desk review team composition.** The desk analysis was conducted by a multidisciplinary team of six experts with complementary clinical, managerial, and public health expertise in MCI preparedness and response. The team included a Senior Medical Officer and Project Manager- Health advisor for Emergency Preparedness and Response; a Consultant General Surgeon and certified instructor in internationally recognized MCI training programs (including ICRC Mass Casualty Incident Training (MCIT), ICRC Health Emergencies in Large Populations (HELP) course, and WHO Mass Casualty Management); an ICRC Head Nurse with expertise in mass casualty management; an MCI trainer who is also the RHUH Nursing Director and Head of the Public Health Emergency Operations Center (PHEOC); a Quality Officer and Nursing Supervisor from RHUH; and a public health specialist with a Master of Public Health (MPH) in Epidemiology and Biostatistics.

**2.1.2. Document selection.**

a. **Internal Documents Selection**

Internal documents were sourced through direct access to hospital archives and requests from relevant departments. The selected documents include:

1. The most recent version of the Code Orange plan

2. Procedural guidelines and protocols

3. Training manuals and educational materials for staff

4. After-action reports from past MCI or disaster drills

5. Hospital policies and memos on emergency preparedness

These documents were handled with strict confidentiality, and permissions were obtained from relevant authorities for accessing and analyzing internal hospital documents.

b. External Documents

A literature review was conducted using targeted keyword searches related to MCI, including "mass casualty," "MCI," "disaster response," "emergency management," "triage," "mass casualty preparedness," "emergency response plan," "crisis management," "disaster management," and "emergency preparedness." Additional keywords focused on hospital

preparedness, such as "hospital preparedness," "hospital response plan," "emergency protocols," "Code Orange," "hospital emergency plan," "hospital disaster plan," and "healthcare facility readiness."

The search strategy incorporated multiple databases and global resources to ensure comprehensive coverage of MCI and emergency preparedness. Peer-reviewed literature was identified through databases such as PubMed and Google Scholar, including relevant studies, reviews, and articles on disaster preparedness and hospital emergency response. In addition, grey literature sources were consulted to capture practical frameworks and guidelines, including publications from the World Health Organization (WHO), Centers for Disease Control and Prevention (CDC), National Incident Management System (NIMS), and the Association for Professionals in Infection Control and Epidemiology (APIC).

**2.1.3. Data analysis.** The team of experts reviewed the documents to identify gaps and areas requiring enhancement. The evaluation centered on four key aspects: the presence of necessary components and protocols within the document, the clarity of procedures and responsibilities, alignment with established guidelines to ensure adherence to best practices, and the applicability of the plan to current operations. To validate the findings, the identified gaps and weaknesses were cross verified with insights from key stakeholders, including ICRC staff and RHUH department heads. This cross-verification process ensured that the desk review's findings were consistent with practical experiences and observations from those directly involved in emergency preparedness and response.

## 2.2. KAP study methodology

**2.2.1. Design and participants.** A cross-sectional study was conducted among staff at RHUH, with participant recruitment taking place from 01/05/2022 to 30/06/2022. The RHUH hospital employs 1,000 health care workers (HCW) from various professional backgrounds, including medical personnel such as physicians, nurses, pharmacists, laboratory, and radiology technicians, as well as non-medical personnel such as administrative staff, maintenance workers, housekeeper supervisors, security personnel, and information technology (IT) staff. The survey targeted all active staff members, excluding those in managerial positions and employees with inactive contracts during the data collection period. Managerial staff was excluded to avoid potential bias, as their perspectives and experiences might significantly differ from those in non-managerial roles. Employees with inactive contracts were excluded to ensure that the data accurately reflected the current working environment and practices within the hospital.

**2.2.2. Sampling methodology.** The sampling methodology for the study involved a two-step process. First, stratified sampling was used to divide the RHUH staff into two main categories: medical personnel (including physicians, nurses, pharmacists, laboratory, and radiology technicians) and non-medical personnel (including administrative staff, maintenance workers, housekeeper supervisors, security personnel, and IT staff). Second, Sampling within these strata was conducted using a convenience sampling approach, where participants were selected based on their availability and willingness to respond to the survey.

**2.2.3. Ethical consideration.** Ethical approval was sought from the Ethics Review Board (ERB) at the ICRC (OP_CORE 22/00004 – CGB/bap) and the institutional review board of RHUH (18-Jan-2022). Participation was completely voluntary and anonymous, with no incentives provided. A detailed explanation of the study's purpose was presented on the initial page of the online questionnaire. Informed consent was obtained electronically from all participants before their involvement. Participants were required to actively indicate their consent by selecting "Yes" before accessing the questionnaire, and they were informed of their right to withdraw at any time without any repercussions. All collected data were kept strictly confidential and anonymized to protect participants' privacy.

**2.2.4 Data collection.** The questionnaire designed to assess KAP regarding MCI was carefully developed with two distinct versions tailored for medical and non-medical staff. Both versions were crafted following an extensive review of guidelines as well as a thorough analysis of previously conducted studies in similar contexts. Additionally, the questionnaires were made available in both English and Arabic to ensure accessibility and comprehensiveness. The English version has been uploaded as a supplementary file for reference. The survey includes:

1. **Demographic Information**: Collects data on age, gender, profession, department, years of experience, and education.

2. **Staff Experience and Training in MCI**: Gathers information about work shifts, the nature and extent of training received, past involvement in MCI, and the number of such incidents experienced. It also explores staff participation in planning and drafting the MCI plan at RHUH Hospital, including details on the most recent training or drill.

3. **Self-Reported Awareness of MCI and RHUH Preparedness Plan:** This section included questions to evaluate the participants' general awareness of critical concepts and procedures related to MCI and the RHUH Code Orange Plan. The questions were primarily closed-ended, with response options such as "Yes," "No," and "Prefer Not to Answer."

4. **Knowledge of MCI Procedures:** This section comprised multiple-choice questions aimed at evaluating participants' understanding of specific MCI procedures, including the concept of triage, the role of the initial responder, and the characteristics of an MCI. For non-medical staff, only one question about the characteristics of an MCI was included. Respondents were asked to select the correct answer from a list of possible options. Each question had one correct answer, and participants received one point for each correct response. Incorrect answers, "I don't know," and "No answer" options were scored as zero. For the medical staff, the total score for this section was calculated by summing the points for all questions, with higher scores indicating a greater understanding of mass casualty procedures.

5. **Knowledge of Procedures Related to Code Orange Protocols:** Participants were queried about the purpose of Code Orange, the triage types used during normal times and MCI, and the specific procedures for triage and activation of the plan. The survey also covered the location of the Code Orange protocol, the contents of the Code Orange kit, and the communication channels for activation. Additionally, HCWs were asked about their responsibilities related to ER evacuation, contacting the Internal Security Forces, and the closure protocols following a Code Orange. Some questions in this section were tailored specifically for non-medical staff, reflecting their distinct roles in the Code Orange protocol. Each question was designed to gauge awareness and understanding, with correct answers receiving points to assess overall knowledge and readiness.

6. **Attitude toward MCI Training, Planning, and Code Orange Responsibilities:** Participants were queried about their training on the MCI plan, their participation in meetings and planning, and their views on the necessity and reasonableness of the MCI plan and Code Orange responsibilities. Key areas also included the necessity of theoretical and practical MCI training, the frequency and adequacy of training and drills, and the coordination between different staff groups.

7. **Practice Response Efficiency and Communication during the application of the Code Orange**: Key questions of this section addressed whether staff knew whom to communicate with during Code Orange, the accessibility of receiving and sorting areas, the existence of evacuation provisions, and the organization of medical records and admissions. Additionally, respondents rated the efficiency of casualty management areas, the sufficiency of resources for casualty movement, and their overall practice response to MCI. The involvement of support departments like dietary, social work, housekeeping, and security during MCI response was also evaluated. Responses were collected through categorical options and Likert scales, where staff rated various aspects of their preparedness and response from 0 (not applicable/not sufficient) to 5 (fully applicable/sufficient).

8. **Staff perceptions regarding their knowledge and involvement in the MCI plan and Code Orange procedures at the hospital:** This section evaluates staff perceptions regarding their knowledge and involvement in the MCI plan and Code Orange procedures at the hospital. Participants rated the extent to which they believe all staff are well-informed about the MCI plan, the clarity and accessibility of the Code Orange protocol, and their own level of involvement in MCI preparedness. Additionally, respondents assessed the hospital's preparedness for mass casualty events, their personal readiness to respond during an MCI, and the applicability and manageability of the triage system. Ratings were provided on a Likert scale, with mean and standard deviation (SD) used to gauge overall staff perceptions and readiness.

9. **Applicability of Code Orange by Referring to the Beirut Blast**: This section begins by confirming whether the respondent was working at RHUH during the Beirut Blast. It assesses their knowledge of specific roles as documented in the Code Orange protocol during the response. The section investigates the notification methods of the MCI situation, the use of the Code Orange kit, and the organization of the emergency room into zones according to the protocol. It also evaluates the admission department's ability to handle the patient influx and identifies any challenges faced while applying the Code Orange protocol. Finally, respondents rate the overall applicability of the Code Orange protocol during the Beirut Blast on a scale from 0 (not applicable) to 5 (highly applicable).

**2.2.5. Procedure.** Our questionnaire was piloted with 5 HCW to assess the readability of the questions and evaluate the overall feasibility of the study. Feedback from the pilot informed minor adjustments, but the pilot data were excluded from the final analysis. The original questionnaire, specifically developed for this study, was designed to be concise, requiring no more than 15 minutes to complete.

HCW were invited to participate in the online questionnaire, hosted on Microsoft Office surveys, via WhatsApp groups. The invitation included a consent form, and the questionnaire was available in both English and Arabic. To boost participation, reminders were sent at one-week intervals.

**2.2.6. Statistical analysis.** For the descriptive analysis, continuous variables were summarized using means and standard deviations, while categorical variables were presented as proportions and percentages. Knowledge assessments were quantified by assigning points to each question, leading to separate percentage mean scores for medical and non-medical staff. Difference in the knowledge total score between medical and non-medical staff was assessed using independent Sample T-test. Attitude and perception assessments were reported as means and standard deviations, as well as frequencies and percentages. P-value less than 0.05 was considered significant. All statistical analyses were conducted using SPSS version 27.0.

## 3. Results

### 3.1. Desk review results

**3.1.1 Overview of the existing 2016 RHUH Code Orange Plan.** The RHUH Code Orange Plan, developed in 2013 and revised in 2016, was designed to guide the hospital's response to MCI with a primary focus on medical services at the ER. This plan was crafted by the multidisciplinary Policy and Procedure Committee and is structured into eight key sections:

1. Policy Statement: stating that the hospital staff will be trained to manage patient surges exceeding ER capacity, with mandatory monthly Code Orange drills. Activation decisions are made by the ER supervisor or clinical supervisor in consultation with the ER chairman.

2. Purpose: Defines the objectives of the plan, specifically to establish a uniform procedure for managing Code Orange incidents effectively.

3. Scope and Applicability: The plan applies solely to the ER.

4. Responsibilities: Defines the key personnel responsible for coordinating and managing the hospital's response during an MCI.

5. Definitions and Abbreviations: Provides clarity on terminology and abbreviations used throughout the plan.

6. Equipment and Tools: Lists the resources required for effective implementation of the plan, including those found in the Orange Kit.

7. Procedure: Describes the step-by-step actions to be taken when Code Orange is activated, including the procedures for mobilizing the ER team and managing the incident. This is organized as tasks for each key personnel responsible.

8. Flowchart: Offers a visual representation of the emergency response process, detailing the sequence of actions during an MCI.

9. References

   In addition to these core sections, the document includes several annexes:

◦ Contents of the Orange Kit, including essential supplies and equipment.

◦ Details about the Code Orange team and their specific roles.

◦ Emergency extensions and pagers associated with the Code Orange response.

◦ The Code Orange file, which contains critical information and resources for managing incidents.

◦ A pharmacy drug list relevant to Code Orange, including anesthetic drugs.

◦ A map of the ER to aid in efficient navigation and resource allocation.

◦ Diffusion protocols for the distribution of resources and information.

**3.1.2. Findings of the search strategy.** The search strategy identified several critical hospital disaster preparedness tools developed by leading global institutions, which provided valuable insights for assessing the Code Orange plan and strengthening RHUH's emergency preparedness protocols.

◦ WHO's extensive guidelines and technical reports on emergency preparedness, such as the Hospital Emergency Response Checklist and Mass Casualty Management Systems, offered a global perspective on best practices that were highly relevant to RHUH's disaster planning [6].

◦ CDC's guidelines on emergency preparedness, including the Healthcare Preparedness and Response Capabilities and resources on bioterrorism, natural disasters, and pandemic response, were essential in evaluating RHUH's current protocols and preparedness measures [25].

◦ NIMS documentation, including the Incident Command System (ICS) guidelines, provided a structured approach to disaster management. This systematic framework was key in understanding how hospitals can organize and coordinate their response during MCI [26].

◦ APIC's resources, particularly those addressing infection control during MCI, helped in understanding how hospitals can manage surges in patient numbers, especially in situations involving infectious diseases. These guidelines offered vital insights into infection prevention protocols and their integration into disaster response [27].

   Additionally, a 2020 study by Al-Hajj et al. was identified as a reference, offering specific criteria for assessing the Code Orange document and establishing foundational elements for RHUH's MCI plan [28].

**3.1.3. Categorization of Disaster Preparedness Components for the Code Orange Plan.** Using the referenced guidelines and tools, disaster preparedness components were systematically extracted and categorized into four main themes: Preparedness, Incident Command Systems (ICS) & Response, Communication, and Management. This structured framework was used to assess the RHUH Code Orange Plan, identifying various levels of alignment and deviations from established standards. Key components of the plan, detailing areas of adherence and pointing out deviations or gaps are highlighted in S1 Table.

**3.1.4. Main findings.**

◦ **The Preparedness theme** evaluates key areas including Planning & Activation, Surge Capacity, Continuity of Essential Services, Evaluation and Drills, and ER Evacuation Protocols. The 2016 RHUH Code Orange Plan aligns with

WHO standards by defining the responsible team and activation procedures. However, it lacks critical details such as specific alert levels and thresholds for incident escalation. The plan does not address surge capacity or continuity of essential services during emergencies, and while it mandates monthly drills and training, it lacks detail on their evaluation. The absence of ER evacuation protocols further highlights significant gaps in the plan, as noted by CDC and APIC guidelines.

◦ **The ICS & Response theme** includes Command and Control, Triage Protocols, Surveillance, and Activation Procedures & Response. The 2016 RHUH Code Orange Plan addresses command and control by defining roles and responsibilities but lacks a structured action flow and a comprehensive Hospital Disaster Control Command Center plan. Triage protocols are mentioned but lack detail, resulting in unclear management during mass casualty incidents. The surveillance system for casualty tracking is inadequately defined, and the activation procedures are outlined only in terms of personnel responsibilities, lacking detailed activation and alert protocols. These gaps contrast with the comprehensive approaches recommended by WHO, CDC, NIMS, and APIC, which emphasize a clear command structure, adaptable triage protocols, robust surveillance systems, and well-defined activation procedures.

◦ **Communication:** This component focuses on reviewing both internal and external communication systems. The 2016 RHUH Code Orange Plan does not specify alternative arrangements for situations where normal communication systems, such as telephones and pagers, fail or become overloaded. Although the plan includes an annex with "Code Orange" contacts, it falls short in detailing provisions for handling communication breakdowns. While the plan emphasizes internal communication, it lacks a comprehensive strategy for external communication with local authorities and media.

◦ **Management:** The 2016 RHUH Code Orange Plan specifies security roles but fails to detail how to ensure safety and security effectively. While human resource management is largely aligned—with established roles and a Code Orange team—the plan lacks standard procedures for resource allocation and logistics. Attachments include a Code Orange file, pharmacy prescription list, and a Code Orange kit. However, the plan does not adequately address the recovery phase, lacking a comprehensive method for returning to regular operations and restocking inventory supplies. Additionally, the ER map is outdated and requires updating.

 **3.1.5. Main recommendations.** Based on the findings from the desk review and analysis of the RHUH Code Orange plan, several key areas for improvement were identified. To enhance the hospital's overall preparedness and response to MCI, the following recommendations are proposed. These suggestions aim to address existing gaps and ensure a more effective, coordinated, and comprehensive emergency response strategy across all levels of staff.

• Clarify the criteria for different stages of plan activation, ensuring staff understand the activation process.

• Provide detailed explanations of the sequence of activation steps.

• Expand SOPs to address surge capacity and ensure continuity of critical services during MCIs.

• Specify protocols for ER operations, including evacuation and triage, as organized processes, not just tasks.

• Properly document training and drill methods, specifying how they will be applied and assessed.

• Enhance the ICS and response plan by introducing an organized flow of operations, including a clear sequence of events and incident-specific actions.

• Define activation criteria with a comprehensive hospital response strategy.

• Strengthen safety and security measures to ensure the hospital's operations remain functional during a disaster.

• Develop a detailed contingency plan addressing supply chain disruptions and other logistical concerns.

- Incorporate a recovery phase in the plan to restore regular operations post-MCI, including service restoration and inventory replenishment.

- Ensure all RHUH staff are informed about the Code Orange Plan, emphasizing preparedness for unexpected MCIs.

- Engage management and key personnel through presentations, internal communications, and workshops to collaboratively identify gaps and prioritize improvements.

The desk review was shared with RHUH management to facilitate the update of the Code Orange Plan through detailed presentations, internal communications, and workshops, ensuring that all relevant stakeholders are informed and engaged. By involving management and key personnel, we aimed to collaboratively identify gaps, prioritize improvements, and implement necessary changes effectively.

### 3.1.6. Update of the Code Orange Plan.
Following the desk review and recommendations, significant changes were made to the RHUH Code Orange Plan to enhance the hospital's preparedness for MCI.

**Key Changes and Actions Taken:**

◦ Restructured Procedure Section: The procedure section has been reorganized into more action-oriented segments, with a defined sequence of events for responding to MCI. This new structure ensures that actions are clearly outlined, improving the efficiency and clarity of emergency responses.

◦ Clarified Command Structure and Notification Process: The command structure framework and notification process have been explicitly defined and clearly presented in the updated plan. This change ensures that roles and responsibilities during an MCI are well understood, leading to better coordination and communication across all teams.

◦ Expanded Safety, Triage, and Activation Procedures: Additional procedural actions have been introduced for safety, triage, and plan activation. These changes include detailed instructions on how to ensure safety during an MCI, a more organized approach to plan activation, and specific guidance on triage. Triage procedures now specify the type of triage to be used, with a visual representation of the integrated triage tool, based on the WHO and ICRC surge tool, included in the plan. This provides clear, standardized criteria for handling large numbers of casualties.

◦ Systematized MCI Phases: The different phases of an MCI are now more systematically defined and structured. This allows for a better-organized response, with clear transitions between the stages of incident management.

◦ Post-Event Debriefing Section: A new section on post-event procedures has been added, outlining how debriefings will be conducted. This section ensures that lessons learned from each MCI can be effectively incorporated into future preparedness and response strategies.

Despite these improvements, some areas still require further attention. The ER map remains outdated and should be revised to reflect the current layout and resources of the emergency department. Additionally, further details on resource management, as well as comprehensive plans for regular drills and training, need to be incorporated into the plan to ensure that staff are well-prepared for real-life scenarios.

### 3.2. KAP Study results

#### 3.2.1. Baseline characteristics of the study participants.
The study sample included 100 hospital staff, of whom 66% were medical and 34% non-medical. Overall, 54% were female and 51% were aged 36–64 years. Educational attainment differed substantially between groups, with 44% of the total sample having ≤12 years of education, compared with 27.3% of medical staff and 76.5% of non-medical staff. More than 10 years of work experience was reported by 60.6% of medical staff versus 29.4% of non-medical staff.

Regarding MCI training, 51% of participants reported no prior training. Practical or drill-based training was reported by 30% overall and was more common among non-medical staff (50%) than medical staff (19.7%). Previous MCI experience was reported by 41% of participants, with higher exposure among medical staff compared to non-medical staff (50% vs. 23.5%). The Beirut Blast and the Iranian Embassy explosion were the most frequently cited incidents. Most participants (78%) reported no involvement in MCI planning, particularly among non-medical staff (85.3% vs. 74.2% of medical staff), while only 15% reported being involved, predominantly medical staff (19.7%) (Table 1).

**3.2.2. Medical Staff Self-Reported Awareness of MCI and RHUH Preparedness plan.** Fig 1 displays self-reported awareness of MCI and RHUH Preparedness plan of the medical and non-medical staff at RHUH. The survey results indicate that RHUH medical staff generally had a high level of awareness of basic MCI concepts, with 77% understanding what an MCI is. A significant majority (85%) were also aware of the Code Orange plan. However, gaps remain, particularly in preparedness and training; only 21% were aware of recent updates to the Code Orange plan, and just 21% believed

**Table 1. Baseline characteristics of the study sample (N = 100).**

| | All (N = 100) | Medical Staff (n = 66) | Non-medical Staff (n = 34) |
|---|---|---|---|
| **Gender n (%)** | | | |
| Male | 46(46%) | 27(40.9) | 19(55.9) |
| Female | 54 (54%) | 39(59.1) | 15(44.1) |
| **Age Category n (%)** | | | |
| 18-25 years | 19(19) | 5(7.6) | 14(41.2) |
| 26-35 years | 30(30) | 24(36.4) | 6(17.6) |
| 36-64 years | 51(51) | 37(56.1) | 14(41.2) |
| **Education level n (%)** | | | |
| ≤12 years | 44(44) | 18(27.3) | 26(76.5) |
| >12 years | 56(56) | 48(72.7) | 8(23.5) |
| **Work experience** | | | |
| 0-1 year | 18(18) | 3(4.5) | 15(44.1) |
| 2-5 years | 18(8) | 11(16.7) | 8(23.5) |
| 5-10 years | 16(16) | 12(18.2) | 1(2.9) |
| More than 10 years | 48(48) | 40(60.6) | 10(29.4) |
| **Training related to MCI** | | | |
| No | 51(51) | 36(54.5) | 15(44.1) |
| Theoretical F2F | 10(10) | 9(13.6) | 1(2.9) |
| Theoretical virtual/online | 5(5) | 5(7.6) | 0(0) |
| Practical/drill | 30(30) | 13(19.7) | 17(50) |
| Practical/tabletop | 4(4) | 3(4.5) | 1(2.9) |
| **Previous work in MCI experience** | | | |
| No | 57(57) | 32(48.5) | 25(73.5) |
| Yes | 41(41) | 33(50) | 8(23.5) |
| No answer | 2(2) | 1(1.5) | 1(2.9) |
| **Involvement in the planning of MCI** | | | |
| No | 78(78) | 49(74.2) | 29(85.3) |
| Yes | 15(15) | 13(19.7) | 2(5.9) |
| No answer | 7(7) | 4(6.1) | 3(8.8) |

N, n frequency, % percentage

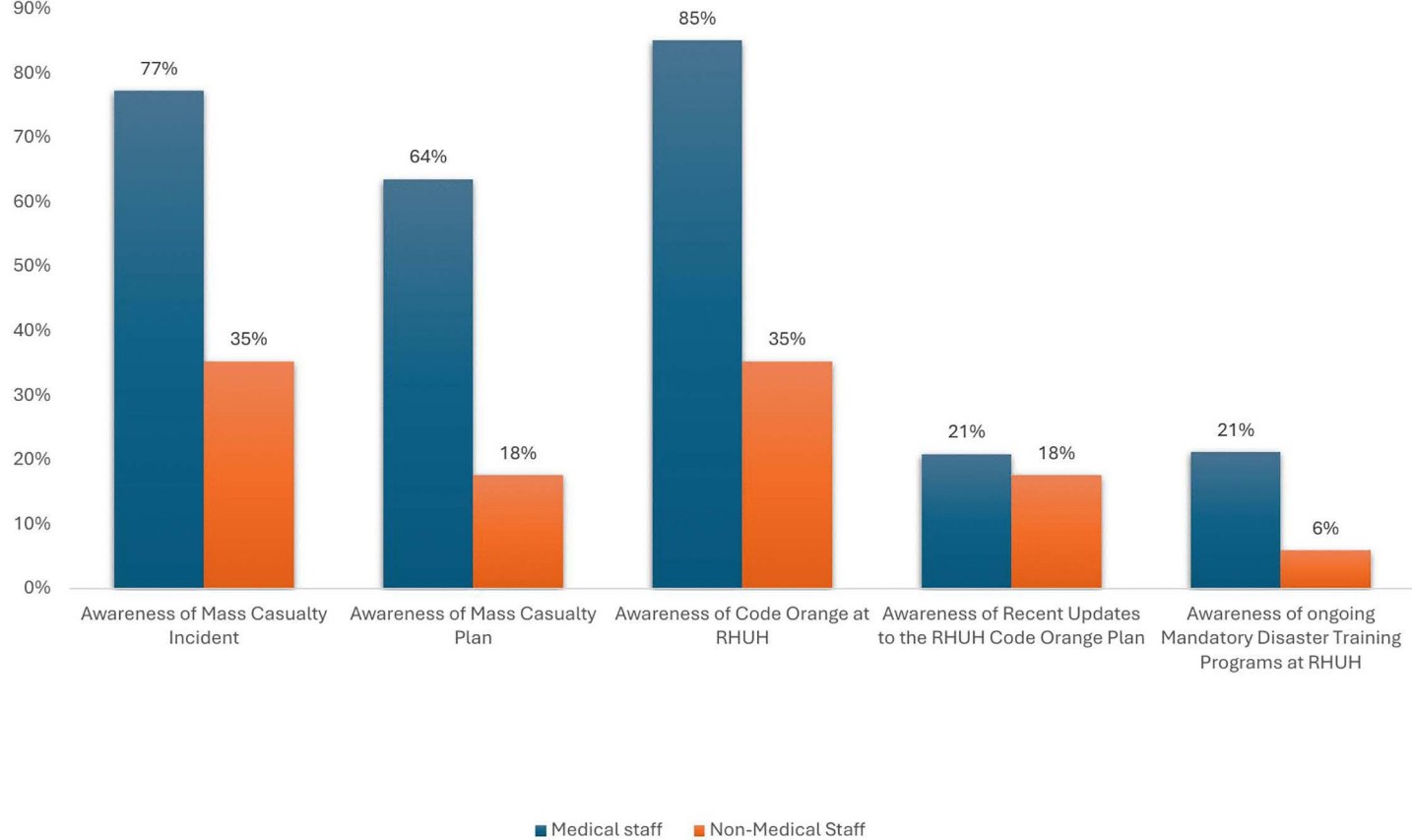

**Fig 1. Awareness Levels of MCI and Code Orange Among Medical and Non-Medical Staff.**

that ongoing mandatory disaster training programs exist. In contrast, non-medical RHUH staff showed low awareness across key areas. Only 35% of the participants knew what an MC, and the same percentage were aware of the Code Orange plan. Awareness of the mass casualty plan, designated areas during triage, and recent updates to the Code Orange plan were even lower, at 18%. Additionally, only 6% were aware of ongoing mandatory disaster training programs.

**3.2.3. Knowledge and practical procedures concerning MCI, triage, and the specific Code Orange protocols at RHUH.** Table 2 shows notable gaps in knowledge of MCI and Code Orange protocols among RHUH staff. Among medical staff, correct identification of triage was high (83.3%); however, knowledge of the initial responder's role (59.1%) and appropriate recipients of first aid (50.0%) was lower. Awareness of MCI characteristics was limited, with a mean MCI knowledge score of 2.5 (SD = 1.18) (out of 4). Knowledge related to Code Orange was substantially lower: 27.9% correctly identified its overall purpose, 7.0% knew the type of triage used during MCI, no one identified who activates Code Orange, and only 1.5% knew where to locate the protocol. The mean Code Orange knowledge score among medical staff was 3.9 (SD = 1.6).

Among non-medical staff, only 35.3% correctly identified MCI characteristics, and 8.8% knew the purpose of triage during Code Orange. No respondents correctly identified the location of Code Orange protocols, 2.9% knew the contents of the Code Orange kit, and 17.6% identified the emergency number for activation. The mean overall knowledge score for non-medical staff was 0.9 (SD = 1.5).

Table 2. Knowledge and practical procedures concerning MCI, triage, and the specific Code Orange protocols at RHUH.

| | Medical Staff | | | Non-Medical Staff | | | |
|---|---|---|---|---|---|---|---|
| | Correct answer | Wrong answer | No answer | Correct answer | Wrong answer | No answer | P-value |
| **Theoretical knowledge about MCI** | | | | | | | |
| What is Triage? * | 55(83.3) | 6(9.1) | 5(7.6) | | | | |
| Initial Responder Actions at MCI Scene* | 39(59.1) | 23(34.8) | 4(6.1) | | | | |
| First Aid Protocols during MCI* | 33(50.0) | 29(43.9) | 4(6.1) | | | | |
| Characteristics of an MCI | 37(56.1) | 21(31.8) | 8(12.1) | 12(35.3) | 20(58.8) | 2(5.9) | <0.001 |
| **MCI Knowledge score Mean (SD)** | 2.5(1.18) | | | | | | |
| **Knowledge about Code Orange Protocols and Procedures at RHUH** | | | | | | | |
| Purpose of Code Orange* | 19(27.9) | 49(72.1) | 0(0) | | | | |
| Triage Type Used at RHUH During Normal Times and MCI* | 15(22.7) | 48(72.7) | 3(4.5) | | | | |
| Triage Type Used During MCI* | 7(7) | 55(83.3) | 4(6.1) | | | | |
| Purpose of Triage during Code Orange | 49(74.2) | 16(24.2) | 1(1.5) | 3(8.8) | 9(26.5) | 22(64.7) | |
| Location of Code Orange Protocol | 1(1.5) | 65(98.5) | 0(0) | 0(0) | 11(32.4) | 23(67.6) | |
| Activation of Code Orange Plan at RHUH | 23(34.8) | 41(62.1) | 2(3) | 6(17.6) | 4(11.8) | 24(70.6) | <0.001 |
| Responsible Party for Activating Code Orange at RHUH | 0(0) | 65(98.5) | 1(1.5) | 0(0) | 11(32.4) | 23(67.6) | |
| Contents of the Code Orange Kit | 38(57.6) | 28(42.4) | 0(0) | | | | |
| Modes of Channels for Activating Code Orange | 12(18.2) | 54(81.8) | 0(0) | 7(20.6) | 5(14.7) | 22(64.7) | |
| Responsibility for ER Evacuation | 13(19.7) | 43(65.2) | 10(15.2) | 2(5.9) | 4(11.8) | 28(82.4) | |
| Responsibility for Contacting ISF During MCI | 7(10.6) | 59(89.4) | 0(0) | 2(5.9) | 10(29.4) | 22(64.7) | |
| Closure Protocols According to Code Orange Documentation | 8(12.1) | 54(81.8) | 4(6.1) | 0(0) | 11(32.4) | 23(67.6) | |
| Emergency Number for Activating Code Orange | 30(45.5) | 30(45.5) | 6(9.1) | 6(17.6) | 6(17.6) | 22(64.7) | |
| **Code Orange Knowledge Score Mean (SD)** | 3.3(2.1) | | | 0.9(1.5) | | | <0.001 |

n Frequency, % Percentage, * for medical staff only, P-value less than 0.05 is considered significant. Top of Form

**3.2.4. Attitude toward MCI Training, Planning, and Code Orange Responsibilities.** Table 3 summarizes attitudes of medical and non-medical staff toward MCI training, planning, and Code Orange responsibilities. Agreement that all staff should be trained on the MCI plan was reported by 86.4% of medical staff and 79.4% of non-medical staff. The need for greater involvement in MCI planning and training was supported by 90.9% of medical staff and 52.9% of non-medical staff. Regular updates of the MCI plan were considered necessary by 86.4% of medical staff, while 58.8% of non-medical staff disagreed. The MCI plan was considered essential by 95.5% of medical staff and 50.0% of non-medical staff. Regarding Code Orange responsibilities, 56.1% of medical staff reported that their roles were reasonable.

Medical staff rated the necessity of theoretical MCI training with a mean score of 2.6 (SD = 0.8). The importance of training frequency scored 4.3 (SD = 0.9). Practical drills were rated at a mean of 4.3 (SD = 0.9), and the frequency of these drills scored 4.2 (SD = 0.9). Coordination between medical and non-medical staff during MCI planning received a mean score of 4.3 (SD = 0.9), while the need for more drills to apply the MCI plan was rated 4.3 (SD = 0.8). Among non-medical staff, the necessity of theoretical MCI training had a mean score of 1.7 (SD = 1.0). Training frequency scored 2.7 (SD = 1.8). Practical drills were rated 2.5 (SD = 1.7), and their frequency scored 2.6 (SD = 1.7). Coordination between staff received a mean score of 2.9 (SD = 1.8), and the need for more drills scored 2.6 (SD = 1.8).

**3.2.5. Practice Response Efficiency and Communication during the Application of the Code Orange.** Among medical staff, 63.6% reported knowing whom to communicate with during Code Orange, and 48.5% indicated that the receiving and sorting areas were accessible and in close proximity to definitive care areas. Additionally, 60.6% of medical

Table 3. Attitudes toward MCI Training, Planning, and Code Orange Responsibilities.

| | Response Options | Medical staff (n=66) | Non-medical staff (n=34) |
|---|---|---|---|
| **Training on MCI Plan n (%)** | No | 8(12.1) | 7(20.6) |
| | Yes, All Staff | 57(86.4) | 27(79.4) |
| | No answer | 1(1.5) | 0(0) |
| **Meeting on MCI Preparedness n (%)** | No | 2(3) | |
| | Yes | 21(31.8) | |
| | No answer | 43(65.2) | |
| **Involvement in MCI Planning and Training * n (%)** | No | 3(4.5) | 15(44.1) |
| | Yes | 60(90.9) | 18(52.9) |
| | No answer | 3(4.5) | 1(2.9) |
| **Updating the MCI Plan n (%)** | No | 5(7.6) | 20(58.8) |
| | Yes | 57(86.4) | 14(41.2) |
| | No answer | 4(6.1) | 0(0) |
| **Necessity of a Mass Casualty Plan n (%)** | No | 2(3) | 16(47.1) |
| | Yes | 63(95.5) | 17(50) |
| | No answer | 1(1.5) | 1(2.9) |
| **Reasonableness of Code Orange Responsibilities * n (%)** | No | 14(21.2) | |
| | Yes | 37(56.1) | |
| | No answer | 15(22.7) | |
| **Necessity of Theoretical MCI Training Mean (SD)** | Scale From not necessary 0–5 highly necessary | 2.6(0.8) | 1.7(1.0) |
| **Frequency of MCI Training Mean (SD)** | Scale From not necessary 0–5 highly necessary | 4.3(0.9) | 2.7(1.8) |
| **Practical Drills for MCI Response Mean (SD)** | Scale From not necessary 0–5 highly necessary | 4.3(0.9) | 2.5(1.7) |
| **Frequency of Practical Drills Mean (SD)** | Scale from rarely 0–5 more often | 4.2(0.9) | 2.6(1.7) |
| **Coordination Between Staff Groups Mean (SD)** | Scale from not applicable 0–5 highly applicable | 4.3(0.9) | 2.9(1.8) |
| **More Drills for MCI Plan Mean (SD)** | Scale from disagree 0–5 agree | 4.3(0.8) | 2.6(1.8) |

* For medical staff only

staff believed that the medical records and admission departments were organized to handle an influx of casualties. Regarding the sufficiency of equipment, supplies, and apparatus for efficient casualty movement, medical staff gave an average rating of 3.89 out of 5. The security department was perceived as highly involved in the MCI response, with a mean score of 4.3. Meanwhile, non-medical staff reported lower awareness and preparedness levels, with fewer indicating that the receiving areas were accessible (20.6%). The security department also received a high involvement rating from non-medical staff, with a mean score of 4.0. Both groups reported moderate practice response efficiency, with medical staff rating it at 3.5 and non-medical staff at 2.3 out of 5 (Table 4).

### 3.2.6. Perceptions of Staff Regarding MCI Preparedness, Code Orange Clarity, and Role Applicability in the Hospital.
Table 5 summarizes the perceptions of medical and non-medical staff regarding various aspects of MCI preparedness and the Code Orange plan within the hospital. Medical staff reported a mean score of 2.6 for their knowledge of the MCI plan, while non-medical staff reported a mean of 1.7. Both groups rated the clarity and accessibility of the Code Orange plan with mean scores around 3.1 and 3.2, respectively. Involvement in MCI preparedness was reflected by mean scores of 3.7 for medical staff and 3.5 for non-medical staff. The hospital's preparedness for MCI through Code Orange received mean scores of 3.6 from medical staff and 3.8 from non-medical staff. Both groups

**Table 4. Medical and Non-Medical Staff Practice Response Efficiency and Communication during the Application of the Code Orange.**

| | Response Options | Medical staff (n = 66) | Non-medical staff (n = 34) |
|---|---|---|---|
| **Do you know whom you communicate when applying code orange? n (%)** | Yes | 42(63.6) | 6(17.6) |
| | No | 20(30.3) | 26(76.5) |
| | Prefer not to answer | 4(6.1) | 2(5.9) |
| **Is the receiving and sorting area accessible and in close proximity to the areas of the hospital in which definitive care will be given? n (%)** | Yes | 32(48.5) | 7(20.6) |
| | No | 26(39.4) | 25(73.5) |
| | Prefer not to answer | 8(12.1) | 2(5.9) |
| **Is there a provision been made for the movement of patients and staff to an immediate area of safe refuge within the hospital in the event the area must be evacuated or staff and patients relocated? n (%)** | Yes | 22(33.3) | 4(11.8) |
| | No | 37(56.1) | 28(82.4) |
| | Prefer not to answer | 7(10.6) | 2(5.9) |
| **Are the medical records and admission departments organized to handle an influx of casualties? * n (%)** | Yes | 40(60.6) | |
| | No | 13(19.7) | |
| | Prefer not to answer | 13(19.7) | |
| **To what extent does the patients' receiving area allow for retention, segregation, and processing of incoming casualties? Mean (SD)** | not applicable 1–5 applicable | 4(1.3) | 4.7(0.8) |
| **To what extent are there sufficient equipment, supplies, and apparatus available, in an organized manner, to permit prompt and efficient casualty movement? Mean (SD)** | From not sufficient 1–5 sufficient | 3.89(1.4) | 3.7(1.0) |
| **In your opinion to what extent your practice response for MCI is efficient? Mean (SD)** | not efficient 1–5 efficient | 3.5(1.0) | 2.3(1.4) |
| **How much do you think that the dietary department are involved and has a role during MCI response? Mean (SD)** | Rate from 1 to 5 | 2.5(1.5) | 2.7(1.3) |
| **How much do you think that the social worker department are involved and has a role during MCI response? Mean (SD)** | Rate from 1 to 5 | 3.2(1.4) | 3.2(1.6) |
| **How much do you think that the housekeeping department are involved and has a role during MCI response? Mean (SD)** | Rate from 1 to 5 | 3.8(1.3) | 3.2(1.2) |
| **How much do you think that the security department are involved and has a role during MCI response? Mean (SD)** | Rate from 1 to 5 | 4.3(1.3) | 4.0(1.2) |

*Only medical staff, n frequency, % percentage, SD Standard Deviation.

reported readiness to respond during an MCI with mean scores of 3.6 and 3.5, respectively, while the applicability of the triage system was rated 3.3 by medical staff.

**3.2.7. Applicability of the code orange plan during the Beirut Blast.** Out of the total participants, 32 reported that they had been working at RHUH during the Beirut Blast, comprising 27 medical staff and 5 non-medical staff. Of these, 31 were aware of their roles as documented in the Code Orange during the Beirut Blast response. When asked how they were informed that they were in an MCI, 4 participants reported being informed through a phone call or a public address system. One participant mentioned being informed via email, another by being at the ER, and one more through other unspecified means. Only five medical staff reported using the Code Orange Kit during their response. Additionally, only eight reported that the ER was divided into zones according to triage, as per the map documented in the Code Orange protocol. Twenty-nine respondents indicated that the admission department was able to handle the influx of patients. However, two reported challenges in applying the Code Orange plan during the Beirut Blast, mainly due to changed infrastructure and supply deficiencies. When asked to rate the applicability of the Code Orange during the Beirut Blast, the mean score was 3.9, with an SD of 0.5.

**Table 5. Perceptions of Staff Regarding MCI Preparedness, Code Orange Clarity, and Role Applicability in the Hospital.**

| | Response Options | Medical staff (n=66) | Non-medical staff (n=34) |
|---|---|---|---|
| To what extent you think all staff are well knowledgeable about MCI plan in the hospital? Mean (SD) | From no knowledge 0–5 expert knowledge | 2.6(0.8) | 1.7(1.0) |
| To what extent do you think that Code Orange is clear and well defined for all staff? Mean (SD) | From not clear 0–5 very clear | 3.1(0.9) | 3.1(1.4) |
| To what extent do you think that Code Orange is accessible to all staff? Mean (SD) | From not accessible 0–5 highly accessible | 3.2(1.1) | 3.2(1.2) |
| To what extent do you feel involved in MCI preparedness in the hospital? Mean (SD) | Scale: From not involved 0–5 highly involved | 3.7(1.0) | 3.5(1.4) |
| To what extent is the hospital prepared for mass casualty by applying Code Orange? Mean (SD) | From not prepared 0–5 highly prepared | 3.6(0.9) | 3.8(0.8) |
| To what extent do you feel that you are ready to respond during MCI? Mean (SD) | From not ready 0–5 ready | 3.6(0.9) | 3.5(1.4) |
| To what extent is the triage used applicable and manageable? * Mean (SD) | From not applicable 0–5 applicable | 3.3(0.8) | |

n frequency, * For medical staff only.

## 4. Discussion

A resilient health system requires effective preparedness and coordinated responses, with the integration of efforts from all involved professionals, particularly healthcare providers. Building on this, the paper offers a comprehensive evaluation of RHUH's MCI preparedness and response by integrating findings from a detailed desk review of the RHUH Code Orange plan and an exploratory analysis of staff KAP related to MCI preparedness and response.

The desk review of the RHUH Code Orange plan revealed both strengths and critical gaps in the hospital's MCI preparedness. While the plan outlines staff roles and available resources for emergency response, it falls short in key areas such as safety, security, resource management, and post-event recovery. Specifically, the plan lacks detailed procedures for crucial aspects like the activation strategy, SOPs for surge capacity, continuity of essential services, and clear triage processes. Additionally, protocols for post-event recovery are either insufficient or absent, and regular drills are not adequately emphasized. Our findings highlight the need for regular revisions to ensure the plan is both comprehensive and actionable. Research from both high- and low-resource settings underscores the necessity of such revisions, emphasizing the importance of having detailed, actionable emergency plans to effectively manage the complexities of mass casualty events. A recent study by Khatri et al. (2023) highlights a comprehensive approach to MCI preparedness, including the integration of public health and primary care, multisectoral coordination, digital tools, and resilient health system planning [29]. The WHO also stresses the need for MCI simulations, which are essential for testing and refining emergency plans while improving staff readiness and communication [4]. Thus, aligning the RHUH Code Orange plan with these best practices will enhance its efficacy in real-world scenarios.

Complementing this analysis, the KAP study assessed the staff's practical knowledge and readiness, revealing notable differences in preparedness between medical and non-medical staff at RHUH. Medical staff showed greater familiarity with MCI concepts and the Code Orange plan, while non-medical staff reported lower awareness and involvement, reflecting a disparity that could hinder overall hospital preparedness. This gap is also evident in their knowledge of recent updates and mandatory training programs, particularly among non-medical staff, who demonstrated limited awareness of ongoing training initiatives and MCI procedures. A similar trend was observed in a study conducted in Poland, which

highlighted the need to adapt training programs based on the specific knowledge gaps of different professional groups, ensuring that tasks are aligned with their respective competencies [30].

Both medical and non-medical staff agreed on the importance of training for MCI preparedness, though medical staff expressed stronger support for involvement in MCI planning and regular updates to the Code Orange plan. This difference underscores a need for better communication and engagement strategies for non-medical personnel, ensuring they are more integrated into the emergency response process. Tailored training, which addresses both groups' specific roles and needs, is critical for fostering a unified response during MCIs.

While medical staff demonstrated higher preparedness and effectiveness in their response, particularly in casualty management and communication, non-medical staff showed lower preparedness and less confidence in their response. This reinforces the need for regular interdisciplinary training and practical drills that involve all staff, emphasizing their respective roles during an MCI. Streamlining training efforts and ensuring the inclusion of non-medical staff in both planning and drills will bridge the gaps in preparedness and response efficiency. Our findings support recommendations from previous studies that emphasize the need for regular, comprehensive training that involves all levels of hospital staff, regardless of their direct involvement in patient care [6,29].

Both groups rated the hospital's overall MCI preparedness similarly, but moderate scores in areas such as readiness and triage applicability highlight room for improvement. The findings emphasize the importance of continuous updates to the Code Orange plan, alongside regular, inclusive training and drills, to ensure all staff members are adequately prepared. This is consistent with existing literature, which emphasizes the need for regular, inclusive training and drills to ensure all staff members are adequately prepared and aware of their roles during an MCI [4,29].

The evaluation of the Code Orange plan's applicability during the Beirut Blast highlights both successes and challenges. While some staff effectively utilized the plan, others faced difficulties related to infrastructure changes and supply shortages, underscoring the need for ongoing refinement of the plan to address real-world scenarios.

### 4.1. Limitations of the study

The desk review was subject to certain limitations. Primarily, the desk review relied solely on existing documents, which may not fully capture the current on-ground practices or informal procedures followed by staff during an MCI. Additionally, given the constraints imposed by the ongoing COVID-19 pandemic, which significantly contributed to the low response rate from hospital staff, the KAP study's results were primarily exploratory in nature. Another limitation is that the study was conducted at a single tertiary care hospital. Despite these challenges, the survey successfully identified critical gaps and training needs in the preparation of medical and non-medical staff for MCI and other disasters. However, a notable limitation of the study was its reliance on self-assessment questionnaires. While this method offers several advantages—such as being inexpensive, practical, fast, scalable, and easy to analyze—there are inherent drawbacks. For instance, respondents may provide socially desirable answers rather than truthful ones, leading to potential biases in the data. To enhance the reliability of future research, it would be beneficial to incorporate more robust data collection methods, such as interviews or focus groups discussion to complement the findings from self-assessment questionnaires.

### 4.2. Recommendations

Based on the findings, it is recommended that RHUH significantly enhanced its MCI preparedness and Code Orange plan through targeted improvements in staff awareness and training. For medical staff, it is crucial to address gaps in knowledge about MCI and Code Orange protocols, focusing on increasing familiarity with their roles and responsibilities. Expanding and updating mandatory disaster training programs with both practical drills and theoretical sessions will bolster preparedness. Non-medical staff require more comprehensive training on MCI procedures and the Code Orange plan to bridge identified knowledge gaps. Additionally, improving communication and clarity around Code Orange procedures,

ensuring all staff are well-informed of updates, and enhancing the accessibility and organization of emergency resources are essential. Regular evaluations and updates to preparedness plans, alongside increased staff involvement in planning and training, will contribute to a more effective response to future incidents. By linking RHUH's need for an updated Code Orange plan with the broader emphasis on continuous education, it becomes clear that revising the plan must be paired with the development of comprehensive training programs. These programs will ensure that healthcare providers are well-prepared and capable of effectively managing MCI, ultimately reinforcing RHUH's role as a central player in Lebanon's disaster response network.

### 4.3. Post-Implementation actions following plan evaluation

Over the two years following the evaluation, RHUH has enacted several improvements, addressing key areas identified in the recommendations:

1. Plan Development and Testing

The MCI plan at RHUH underwent significant updates, with drafting efforts involving collaboration across hospital departments. This inclusive approach, facilitated through multiple bilateral meetings and discussions, ensured that input from all relevant stakeholders informed the final action steps. Prior to implementing the updated plan, RHUH engaged support departments in a series of tabletop exercises, followed by a drill to test readiness. A large-scale drill involving various stakeholders and EMS providers further validated the plan's preparedness.

2. Human Resources and Capacity Building

Staff capacity building has been strengthened through a series of post-incident analyses conducted in partnership with the ICRC. Training sessions included healthcare and non-healthcare staff, enhancing response efficiency across the workforce. Updated role definitions within the incident command team have also improved coordination during emergency responses.

3. Logistics and Equipment

Logistics enhancements prioritized medical and non-medical equipment to ensure the availability of adequate resources during incidents. A comprehensive SOP for patient transfer and referral was also established, covering all aspects of the MCI plan, and outlining clear internal and external roles and responsibilities.

4. Infrastructure and Access Pathways

Hospital infrastructure underwent renovations to facilitate smoother responses during MCIs, particularly by improving access routes and pathways. These changes aim to streamline patient movement and optimize hospital flow during emergency situations.

5. Hospital Resilience

Hospital resilience is strengthened through reinforced physical structures, ongoing maintenance, and the implementation of backup systems for power, water, and communication to ensure uninterrupted operations during emergencies. Designated safe zones within the facility provide secure areas for patients and staff, while a robust stockpiling and supply chain continuity plan ensures that essential resources remain available during MCIs.

6. Emergency Response Preparedness

Emergency preparedness has been integrated into specific objectives, with a focus on implementing comprehensive preparedness and contingency plan. Regular drills, surveys, and updates to the plan ensure that RHUH continuously assesses and improves readiness based on staff knowledge and training outcomes.

## 5. Conclusion

In conclusion, the assessment of RHUH's Code Orange plan and staff preparedness has identified significant gaps that, if addressed, could greatly enhance the hospital's response to MCI. Findings revealed a need for clearer procedures, updated protocols, and stronger communication strategies, particularly among non-medical staff. To address these issues, recommendations include a comprehensive update to the Code Orange plan, regular training programs, and targeted awareness initiatives to strengthen staff knowledge and coordination. Future directions for sustaining these improvements involve implementing robust monitoring and evaluation systems, engaging key stakeholders, and fostering collaboration with external emergency response agencies such as government agencies, non-governmental organizations (NGOs), ambulance services, and organizations like the Red Cross. By taking these steps, RHUH can build a resilient, well-prepared response system that ensures effective, coordinated action during future emergencies, ultimately leading to improved outcomes for both patients and the community.

Beyond the RHUH context, these findings highlight key considerations for hospitals operating in similar settings. Strengthening MCI preparedness requires not only comprehensive and regularly updated emergency plans, but also inclusive, interdisciplinary training that engages both medical and non-medical staff. Ensuring alignment between planning, training, and real-world implementation—supported by continuous evaluation and simulation exercises—can serve as a practical framework for improving hospital resilience and emergency response capacity in resource-constrained and crisis-prone environments.

## Supporting information

**S1 Table. Summary of Staff Suggestions for Improving Retention at RHUH.**
(DOCX)

## Acknowledgments

The authors would like to thank all the health care workers at RHUH who accepted to participate in the study for their time and cooperation.

## Author contributions

**Conceptualization:** Rima Kashash, Mustapha Khalife, Mohamad Shafic Ramadan.

**Data curation:** Linda Abou-Abbas, Rima Kashash.

**Formal analysis:** Linda Abou-Abbas, Rima Kashash.

**Methodology:** Linda Abou-Abbas, Rima Kashash, Mustapha Khalife, Mohamad Shafic Ramadan.

**Project administration:** Mustapha Khalife.

**Software:** Linda Abou-Abbas.

**Supervision:** Rima Kashash, Mustapha Khalife, Mohamad Shafic Ramadan.

**Validation:** Linda Abou-Abbas.

**Visualization:** Linda Abou-Abbas, Mustapha Khalife, Mohamad Shafic Ramadan.

**Writing – original draft:** Linda Abou-Abbas, Rima Kashash.

**Writing – review & editing:** Mustapha Khalife, Mohamad Shafic Ramadan.

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
