## [Decision Letter · Decision Letter 0]

17 Mar 2026

Dear Dr. Abou-Abbas,

Thank you for submitting your manuscript to PLOS ONE. After careful consideration, we feel that it has merit but does not fully meet PLOS ONE’s publication criteria as it currently stands. Therefore, we invite you to submit a revised version of the manuscript that addresses the points raised during the review process.

It is really a well written study dealing with a extremely important topic. Both reviewers recommend a minor revision only prior to definite acceptance. Please, clarify the statistics methods, as pointed out by reviewer 1. Reviewer 2 asked about a possible error in lines 521. I think, it will be quite simple to make any correction.

We look forward to receiving your revised manuscript. Once again, your important manuscript deserves widespread distribution, as it deals with a very important topic.

Kind regards,

Hans-Peter Simmen, M.D., Professor of Surgery

Academic Editor

PLOS One

Journal Requirements:

2. In your ethics statement, please ensure you have provided the full name of the ethics committee(s) that approved this study. If more than 5 committees provided approval, it is acceptable to provide a list as a Supporting Information file.

4. In the online submission form you indicate that your data is not available for proprietary reasons and have provided a contact point for accessing this data. Please note that your current contact point is a co-author on this manuscript. According to our Data Policy, the contact point must not be an author on the manuscript and must be an institutional contact, ideally not an individual. Please revise your data statement to a non-author institutional point of contact, such as a data access or ethics committee, and send this to us via return email. Please also include contact information for the third party organization, and please include the full citation of where the data can be found.

5. Please amend either the title on the online submission form (via Edit Submission) or the title in the manuscript so that they are identical.

Reviewers' comments:

Reviewer's Responses to Questions

**Comments to the Author**

1. Is the manuscript technically sound, and do the data support the conclusions?

Reviewer #1: Yes

Reviewer #2: Yes

2. Has the statistical analysis been performed appropriately and rigorously?

Reviewer #1: No

Reviewer #2: I Don't Know

3. Have the authors made all data underlying the findings in their manuscript fully available?

Reviewer #1: No

Reviewer #2: Yes

4. Is the manuscript presented in an intelligible fashion and written in standard English?

Reviewer #1: Yes

Reviewer #2: Yes

Reviewer #1: Thank you for the opportunity to review this interesting paper. The manuscript addresses an important and relevant topic. Focus of this manuscript is the evaluation of the mass casualty preparedness of the RHUH hospital via a desk review of the organizations Code Orange plan and an evaluation of staff knowledge via a cross-sectional KAP survey regarding MCI preparedness.

Methods:

Desk Review:

The "team of experts" conducting the desk analysis shoud be described more clearly. How many experts? Which qualifications?

Statistics:

the manuscript aims to compare differences in percieved preparedness of non medical and medical staff - in the statistics section it is stated that a paired t-test was used for assessment. Please clarify whether an independent samples t-test was performed instead, as the groups appear to be independent? If a paired test was used, the statistical approach should be reconsidered.

Conclusion:

Right now the conclusion focuses on measures to improve the disaster preparedness at the RHUH - the authors should consider expanding the conclusion to include more generizable recomendations. Thus, it could be read more as a potential blueprint for other hospitals and provide a broader perspective, strengthening relevance of the findings.

Figure 1:

should be uploaded in better resolution, the text is not readable. A figure legend should be added.

Other:

Line 218: a ) is missing after IT staff

Reviewer #2: Thank you for the opportunity to review this study. This study conducted a desk review and a survey of hospital staff in regard to the mass casualty contingency plan for a large tertiary government hospital in Lebanon. The paper highlights many improvements that can be made and that have been implemented that are important for all medical institutions worldwide to address. The survey well highlights the difficulty associated with staff awareness and training, in what is already a challenging and very strenuous work environment.

How do we efficiently prepare staff for an MCI when resources in both time and money are limited? It’s a difficult question to answer, and this paper does a good job of addressing the difficulties inherent in planning for an MCI, which are of course exceptionally taxing and challenging. Further, it provides helpful inputs on where to focus energies and how to be better prepared.

I would like to see two questions answered in the paper:

1) who specifically, performed the desk review?

2) was a debriefing performed after the Beirut Blast? How many patients did RHUH treat during this terrible explosion? What was learned from that event and how has this been applied to the current MCI plan? This would be very interesting to learn about, as the facility has the rare experience of handling a large MCI. Why did only 8 of 32 staff members use the code orange kit, for example?

Addressing these questions will greatly enhance the paper as it provides actual data on MCI management. The previous MCI plan was revised in 2016. The Beirut Blast occurred in 2020. So it would be of great interest to discuss how the update to the current MCI contingency plan was influenced by a major real-world MCI. This should be a critical strength of this paper!

There is an error on line 521: more medical staff had previous MCI experience than non-medical staff. On line 521, the rates of no previous exposure have erronously been reported.

I would very briefly highlight, that the purpose of improving such protocols and procedures is not just ensuring safety for hospital staff, but for the benefit of patient care, this is lacking in the abstract, although it is well addressed in the introduction.

Finally, the paper is well-written and requires no revision in this regard.

Overall, this is a well performed study that provides valuable insights highlighting the importance of properly informing and training hospital staff about MCI contingency plans, so that staff are safe and as effective as possible in providing patient care. I thank the authors for their efforts and recommend that the study be accepted for publication with minor revisions. I would recommend strengthening the paper by highlighting the role that actual real-world experience in dealing with the Beirut Blast (and also Iranian Embassy Explosion) helped improve the current MCI protocol.

.

Reviewer #1: No

Reviewer #2: No

---

## [Author Response · Author response to Decision Letter 1]

7 Apr 2026

Reviewer #1:

Methods:

Desk Review:

The "team of experts" conducting the desk analysis should be described more clearly. How many experts? Which qualifications?

Author’s reply: We have revised the manuscript to provide a clearer description of the “team of experts” involved in the desk analysis. Specifically, we now indicate the number of experts (six) and detail their respective qualifications and roles, highlighting their multidisciplinary expertise in clinical care, nursing leadership, emergency preparedness, and public health.

Statistics:

the manuscript aims to compare differences in perceived preparedness of non-medical and medical staff - in the statistics section it is stated that a paired t-test was used for assessment. Please clarify whether an independent samples t-test was performed instead, as the groups appear to be independent? If a paired test was used, the statistical approach should be reconsidered.

Author’s reply: Thank you for highlighting this point. The analysis was conducted using an independent samples t-test, as the comparison involved two independent groups (medical vs. non-medical staff). The mention of a paired t-test in the Methods section was a reporting error and has now been corrected.

Conclusion:

Right now the conclusion focuses on measures to improve disaster preparedness at the RHUH - the authors should consider expanding the conclusion to include more generalizable recommendations. Thus, it could be read more as a potential blueprint for other hospitals and provide a broader perspective, strengthening relevance of the findings.

Author’s reply: Thank you for this valuable suggestion. The conclusion has been revised to include more generalizable recommendations beyond the RHUH context. Specifically, we have expanded the section to highlight key considerations for hospitals operating in similar settings, emphasizing the importance of comprehensive emergency planning, inclusive interdisciplinary training, and alignment between planning, training, and real-world implementation.

Figure 1:

should be uploaded in better resolution; the text is not readable. A figure legend should be added.

Author’s reply: The figure has been uploaded in a higher resolution, as recommended by the journal. The figure legend is provided within the manuscript (Line 560).

Other:

Line 218: a) is missing after IT staff

Author’s reply: Corrected.

Reviewer #2: Thank you for the opportunity to review this study. This study conducted a desk review and a survey of hospital staff in regard to the mass casualty contingency plan for a large tertiary government hospital in Lebanon. The paper highlights many improvements that can be made and that have been implemented that are important for all medical institutions worldwide to address. The survey well highlights the difficulty associated with staff awareness and training, in what is already a challenging and very strenuous work environment. How do we efficiently prepare staff for an MCI when resources in both time and money are limited? It’s a difficult question to answer, and this paper does a good job of addressing the difficulties inherent in planning for an MCI, which are of course exceptionally taxing and challenging. Further, it provides helpful inputs on where to focus energies and how to be better prepared.

I would like to see two questions answered in the paper:

1) who specifically performed the desk review?

Author’s reply: We have revised the manuscript to provide a clearer description of the “team of experts” involved in the desk review. Specifically, we now indicate the number of experts (six) and detail their respective qualifications and roles, highlighting their multidisciplinary expertise in clinical care, nursing leadership, emergency preparedness, and public health.

2) was a debriefing performed after the Beirut Blast? How many patients did RHUH treat during this terrible explosion? What was learned from that event and how has this been applied to the current MCI plan? This would be very interesting to learn about, as the facility has the rare experience of handling a large MCI.

Author’s reply: Thank you for this valuable comment. We have revised the Introduction to address these points. The manuscript has been updated accordingly:

To strengthen preparedness for MCIs, RHUH established its mass casualty response plan, Code Orange, in 2016. The plan provides a structured framework for hospital response during large-scale emergencies, including activation criteria, triage processes, staff mobilization, and surge capacity management. RHUH has applied this plan in response to several large-scale events, most notably the Beirut Port explosion in 2020, during which approximately 220 casualties presented to the hospital, with 48 requiring admissions, including critical care. This event placed unprecedented strain on the hospital and provided critical insights into both the strengths and limitations of existing response mechanisms. Following the explosion, a formal debriefing was conducted to inform updates to the Code Orange plan and incorporate lessons learned from this real-world experience. Identified challenges—such as rapid patient influx, interdepartmental coordination, and resource limitations—led to targeted improvements in triage systems, surge capacity, and staff mobilization protocols, ensuring a more practical and adaptive response to large-scale MCIs.

There is an error on line 521: more medical staff had previous MCI experience than non-medical staff. On line 521, the rates of no previous exposure have erroneously been reported.

Author’s reply: Thank you for pointing out this issue. Corrected

I would very briefly highlight, that the purpose of improving such protocols and procedures is not just ensuring safety for hospital staff, but for the benefit of patient care, this is lacking in the abstract, although it is well addressed in the introduction.

Author’s reply: Added please refer to the abstract

Finally, the paper is well-written and requires no revision in this regard. Overall, this is a well performed study that provides valuable insights highlighting the importance of properly informing and training hospital staff about MCI contingency plans, so that staff are safe and as effective as possible in providing patient care. I thank the authors for their efforts and recommend that the study be accepted for publication with minor revisions.

Author’s reply: We sincerely thank the reviewer for this positive feedback and are pleased that the manuscript is considered well-written. We appreciate your time and thoughtful evaluation of our work.

I would recommend strengthening the paper by highlighting the role that actual real-world experience in dealing with the Beirut Blast (and also Iranian Embassy Explosion) helped improve the current MCI protocol.

Author’s reply: Thank you for your comment. We have revised the introduction to highlight how real-world experience from the Beirut Port explosion directly informed improvements to the Code Orange plan. The updated paragraph now emphasizes the formal debriefing, lessons learned, and how these insights were translated into practical updates to triage processes, surge capacity, and staff mobilization protocols.

We believe this revision clearly addresses your suggestion and strengthens the link between operational experience and protocol enhancement.

---

## [Editor Report · Decision Letter 1]

14 Apr 2026

Mass Casualty Incident Preparedness and Response: A Desk Review of the Code Orange Plan and Assessment of Healthcare Workers' Knowledge, Attitudes, and Practices in a Lebanese Tertiary Government Hospital

PONE-D-26-02078R1

Dear Dr. Abou-Abbas,

We’re pleased to inform you that your manuscript has been judged scientifically suitable for publication and will be formally accepted for publication once it meets all outstanding technical requirements. Following minor revision your manuscript is really much better as well as more informative. Congratulations.

Kind regards,

Hans-Peter Simmen, M.D., Professor of Surgery

Academic Editor

PLOS One
---

## [Editor Report · Acceptance letter]

PONE-D-26-02078R1

PLOS One

Dear Dr. AbouAbbas,

I'm pleased to inform you that your manuscript has been deemed suitable for publication in PLOS One. Congratulations! Your manuscript is now being handed over to our production team.

Kind regards,

on behalf of

Dr. Hans-Peter Simmen

Academic Editor

PLOS One